# Balloon-Expandable versus Self-Expandable Valves in Transcatheter Aortic Valve Implantation: Complications and Outcomes from a Large International Patient Cohort

**DOI:** 10.3390/jcm10174005

**Published:** 2021-09-04

**Authors:** Astrid C. van Nieuwkerk, Raquel B Santos, Leire Andraka, Didier Tchetche, Fabio S. de Brito, Marco Barbanti, Ran Kornowski, Azeem Latib, Augusto D’Onofrio, Flavio Ribichini, Francisco Ten, Nicolas Dumonteil, Jan Baan, Jan J. Piek, Alexandre Abizaid, Samantha Sartori, Paola D’Errigo, Giuseppe Tarantini, Mattia Lunardi, Katia Orvin, Matteo Pagnesi, Juan Manuel Nogales-Asensio, Angie Ghattas, George Dangas, Roxana Mehran, Ronak Delewi

**Affiliations:** 1Heart Center, Amsterdam UMC, Department of Cardiology, University of Amsterdam, Amsterdam Cardiovascular Sciences, Meibergdreef 9, 1105 AZ Amsterdam, The Netherlands; A.vannieuwkerk@amsterdamumc.nl (A.C.v.N.); raquelsantos.cardiologia@chporto.min-saude.pt (R.B.S.); j.baan@amsterdamumc.nl (J.B.); j.j.piek@amsterdamumc.nl (J.J.P.); 2Department of Cardiology, Centro Hospitalar Universitário do Porto, Largo Prof. Abel Salazar, 4099-001 Porto, Portugal; 3Hospital de Basurto, 48013 Bilbao, Spain; Leire.andrakaikazuriaga@osakidetza.eus; 4Clinique Pasteur, 31300 Toulouse, France; d.tchetche@clinique-pasteur.com (D.T.); ndumonteil@clinique-pasteur.com (N.D.); Aghattas@adhb.govt.nz (A.G.); 5Heart Institute, University of São Paulo Medical School, São Paulo 05403-900, Brazil; fabio.cardiol@gmail.com (F.S.d.B.); aabizaid@uol.com.br (A.A.); 6Division of Cardiology, Policlinico-Vittorio Emanuele Hospital, University of Catania, 95123 Catania, Italy; mbarbanti83@gmail.com; 7Rabin Medical Center, Cardiology Department, Petach Tikva 49100, Israel; ran.kornowski@gmail.com (R.K.); katiaorvin@gmail.com (K.O.); 8Division of Cardiology, Department of Medicine, University of Cape Town, Cape Town 7925, South Africa; alatib@gmail.com; 9Montefiore Medical Center, Department of Interventional Cardiology, New York, NY 10467, USA; 10Division of Cardiac Surgery, University of Padova, 35128 Padova, Italy; adonofrio@hotmail.it (A.D.); giuseppe.tarantini.1@unipd.it (G.T.); 11Division of Cardiology, Department of Medicine, University of Verona, 37126 Verona, Italy; flavio.ribichini@univr.it (F.R.); mattia.lunardi@outlook.com (M.L.); 12Servicio de Cardiología, Hospital Universitario y Politécnico La Fe, 46026 Valencia, Spain; paco_ten@yahoo.es; 13The Zena and Michael A. Wiener Cardiovascular Institute, Icahn School of Medicine at Mount Sinai, New York, NY 10029, USA; samantha.sartori@mountsinai.org (S.S.); george.dangas@mountsinai.org (G.D.); roxana.mehran@mountsinai.org (R.M.); 14National Centre for Global Health—Instituto Superiore di Sanità, 00161 Rome, Italy; paola.derrigo@iss.it; 15Institute of Cardiology, ASST Spedali Civili, Department of Medical and Surgical Specialties, Radiological Sciences and Public Health, University of Brescia, 25123 Brescia, Italy; m.pagnesi@gmail.com; 16Servicio de Cardiología, Hospital Universitario de Badajoz, 06006 Badajoz, Spain; juanmanogales@yahoo.es

**Keywords:** aortic valve stenosis, transcatheter aortic valve implantation, mortality, stroke, balloon-expandable, self-expandable

## Abstract

Background: Both balloon-expandable (BE) and self-expandable (SE) valves for transcatheter aortic valve implantation (TAVI) are broadly used in clinical practice. However, adequately powered randomized controlled trials comparing these two valve designs are lacking. Methods: The CENTER-study included 12,381 patients undergoing transfemoral TAVI. Patients undergoing TAVI with a BE-valve (*n* = 4096) were compared to patients undergoing TAVI with an SE-valve (*n* = 4096) after propensity score matching. Clinical outcomes including one-year mortality and stroke rates were assessed. Results: In the matched population of *n* = 5410 patients, the mean age was 81 ± 3 years, 60% was female, and the STS-PROM predicted 30-day mortality was 6.2% (IQR 4.0–12.4). One-year mortality was not different between patients treated with BE- or SE-valves (BE: 16.4% vs. SE: 17.0%, Relative Risk 1.04, 95%CI 0.02–1.21, *p* = 0.57). One-year stroke rates were also comparable (BE: 4.9% vs. SE: 5.3%, RR 1.09, 95%CI 0.86–1.37, *p* = 0.48). Conclusion: This study suggests that one-year mortality and stroke rates were comparable in patients with severe aortic valve stenosis undergoing TAVI with either BE or SE-valves.

## 1. Introduction

Transcatheter aortic valve implantation (TAVI) is a life-saving treatment for patients with severe aortic valve stenosis. Although it was originally developed and indicated for inoperable patients, its treatment area is rapidly expanding to lower risk patients [1,2]. Since its introduction, two different valve designs have been approved and are widely used: balloon-expandable (BE) and self-expandable (SE) valves. BE-valves (Edwards Sapien, Sapien XT, and Sapien 3) are built out of bovine pericardium leaflets on the inside of the cobalt chromium frame. They are expanded during rapid pacing by inflation of a balloon [3]. SE-valves (Medtronic CoreValve and CoreValve Evolut R) are built out of three porcine pericardium leaflets attached to a nitinol self-expanding framework [4]. Despite two clear different mechanisms, clinical guidelines do not favor either of these two different valve designs [5,6]. Previously we have shown similar mortality but lower stroke rates in patients treated with BE-valves at 30-day follow-up [7]. However, data beyond 30 days are scarce. Various small randomized controlled trials have found conflicting results. They were all underpowered to detect superiority of one valve type over the other [8,9,10]. Two recent French observational studies showed higher two-year mortality in patients receiving BE-valves than in patients receiving SE-valves [11,12]. However, it is unknown whether these results reflect only local practice. Given the limited and conflicting data beyond 30 days follow-up, we aimed to assess one-year mortality and stroke rates in patients receiving BE vs. SE-valves.

## 2. Materials and Methods

### 2.1. Study Design and Patient Sample

The Cerebrovascular Events in Patients Undergoing Transcatheter Aortic Valve Implantation (CENTER) study is an international collaboration, including patients with severe aortic valve stenosis undergoing transfemoral TAVI with BE-devices (Edwards Lifesciences Inc., Irvine California, CA, USA) or SE-devices (Medtronic Inc., Minneapolis, MN, USA). The study is registered at clinicaltrials.gov (NCT03588247). A detailed description on study design, eligibility criteria, systematic search methodology, and data collection has been published previously [7]. In short, the study includes data from 3 national registries, 2 multicenter prospective registries, 4 single center prospective registries, and 1 randomized clinical trial identified through a systematic search on PubMed. To ensure sufficient operator experience, studies were included in this analysis if the centers reported at least fifty procedures per valve type. Hence, the study includes patient level data from a global patient sample treated in the United States of America, Brazil, Israel, Spain, Italy, and France. The choice for a BE or SE valve was made by the local heart team of each center. An overview of the included studies is presented in Appendix A. All studies were conducted according to the Declaration of Helsinki and all patients provided written informed consent at each participating center. Collaborators provided a dedicated database with baseline patient characteristics, echocardiographic data, procedural information, and long-term clinical follow-up data. Accordingly, a total of 12,381 patients undergoing transfemoral TAVI between 2007 and 2018 with either BE or SE-valves were included in the CENTER-study.

### 2.2. Study Endpoints

Clinical endpoints of this analysis were differences in all-cause mortality and stroke in patients treated with BE-valves vs. SE-valves occurring within the first year after TAVI. Endpoints were defined by the standardized definitions from the Second Valve Academic Research Consortium (VARC2) [13]. Only the OBSERVANT trial defined stroke as a neurological deficit lasting more than 24 h, or less than 24 h in case of positive neuroimaging, which is equivalent to the VARC2-definition of stroke [14].

We performed different subgroup analyses to compare mortality and stroke outcomes in patients treated with BE vs. SE-valves. First, we performed subgroup analyses by sex. Second, we grouped patients according to valve generation: Early generation valve types (Edwards Sapien, Sapient XT, and Medtronic CoreValve) vs. third-generation valve types (Sapien 3 and Evolut R). Both third-generation valve types were approved in 2014 in Europe and in 2015 in the United States. Third, subgroup analyses per time period (three time periods: 2007–2010, 2011–2014, 2015–2018) were conducted. Fourth, subgroup analyses by valve size were performed. Available valve sizes for BE-valves were 20 mm, 23 mm, 26 mm, and 29 mm. SE-valves were available in 23 mm, 26 mm, 29 mm, 31 mm (CoreValve), and 34 mm (Evolut R). For subgroup analysis, valve size was categorized into three groups: Small (20–23 mm), medium (26 mm), and large (29–34 mm).

### 2.3. Statistical Analysis

Patients treated with BE-valves were compared to patients treated with SE-valves. Baseline characteristics of these two groups were assessed. Continuous variables were tested for normality and then reported as mean and standard deviation, or as median and interquartile ratio (IQR). Missing data were assumed to be missing at random. Therefore, we estimated missing data using multiple imputation methods. The imputation procedure concerning multivariate regression models was performed according to Rubin’s protocol. We applied propensity score matching in order to reduce potential confounding and selection bias. The propensity score was calculated using a logistic regression model including baseline characteristics that either significantly predicted treatment (BE or SE valve) or outcomes (mortality and stroke). Consequently, twelve variables were used to calculate the propensity matching score: Age, sex, body mass index, logistic EuroSCORE, previous myocardial infarction, previous percutaneous coronary intervention, previous stroke or transient ischemic attack (TIA), peripheral artery disease, atrial fibrillation, significant coronary artery disease, dyslipidemia, and renal failure (glomerular filtration ratio of less than 30 mL/min/1.73 m^2^). Each patient treated with a BE-valve was paired with a patient treated with an SE-valve based on the nearest propensity score. Matching was performed using the nearest neighbor method in a 1:1 ratio and no replacement. Within the propensity-matched population, distributions of baseline characteristics were assessed with standardized mean differences. These were calculated for each variable as differences in means or proportions divided by the pooled estimate of the standard deviation. The difference between BE- and SE-valve patients was considered negligible when the standardized mean difference was 0.1 or lower (Appendix A).

Differences in clinical outcomes, mortality, and stroke incidence in both the total and matched cohort were tested using Chi-square. The relative risk (RR) and corresponding asymptotic two-sided 95% confidence interval (CI) was reported. Time-to-event mortality curves were established using cox regression analysis and hazard ratios (HR) were calculated. Early clinical outcomes, according to VARC2, were compared between patients treated with BE- and SE-valves. If *p* < 0.05, the variable was tested in a binary logistic regression model as a potential predictor of one-year mortality and odds ratios (OR) were reported. All statistical tests were two-tailed, and *p* < 0.05 was considered statistically significant. Calculations were performed using SPSS software (version 26.0 for Windows, SPSS, Inc., Chicago, IL, USA).

## 3. Results

### 3.1. Baseline Characteristics of the Overall Study Population

A total of 12,381 patients undergoing TAVI received either a BE- (*n* = 6239) or SE-valve (*n* = 6142). Baseline characteristics of the overall and propensity-matched population are presented in Appendix A. After propensity matching, a total of 8192 patients with either BE- (*n* = 4096) or SE-valves (*n* = 4096) were included. In this matched population, one-year follow-up was completed in 5410 patients. One-year follow-up was completed in 74% of SE-valve patients and 58% of BE-valve patients. Baseline and procedural characteristics of patients with vs. without one-year follow-up completed are presented in Appendix A. Patient flow is presented in Figure 1.

### 3.2. Baseline Characteristics of the Matched Study Population

In the propensity-matched population, 5410 patients completed one-year follow-up and were thus included in the current analysis. Mean age was 81 ± 3 years and 60% were female. The median predicted 30-day mortality with STS-PROM was 6.2% (IQR 4.0–12.4). Table 1 presents baseline and procedural characteristics of this patient population with either BE- or SE-valves. 

Demographics, medical history, and mortality risk prediction scores were comparable. In particular, the presence of coronary artery disease was similar. However, implanted SE-valves had a larger valve size than BE-valves (BE: 26, IQR 23–26, vs. SE: 26, IQR 26–29 mm, *p* < 0.001).

### 3.3. One-Year Mortality

During follow-up (median 365 days, IQR 53–666), 895 (16.8%) patients died in the propensity-matched population. One-year mortality was 16.4% in patients treated with BE-valves and 17.0% in SE-valves (RR 1.04, 95%CI 0.92–1.21, *p* = 0.57). Figure 2 shows Kaplan–Meier survival curves for patients treated with these valve types (HR 0.99, 95%CI 0.87–1.14, *p* = 0.95).

One-year mortality between patients receiving BE- or SE-valves was also comparable in the unmatched population (BE: 17.1% vs. SE: 16.6%, RR 0.97, 95%CI 0.88–1.07 *p* = 0.52).

Mortality was similar across valve type when split for sex: In female patients (BE: 15.9% vs. SE: 15.4%, RR 0.97, 95%CI 0.82–1.14, *p* = 0.70) as well as in male patients (BE: 17.2% vs. SE: 19.4%, RR 1.13, 95%CI 0.94–1.36, *p* = 0.19). In patients treated with third-generation valves (Edwards Sapien 3 and Medtronic CoreValve Evolut R) mortality was comparable: 10.7% in BE-valve recipients vs. 12.7% in SE-valve recipients (RR 1.19, 95%CI 0.89–1.59, *p* = 0.25). Furthermore, in patients treated with older-generation valves, one-year mortality was similar (BE: 18.7% vs. SE: 18.9%, RR 1.01, 95%CI 0.87–1.13, *p* = 0.86).

In the earlier years of TAVI (2007–2010), mortality was comparable between both valve types (BE: 22.2% vs. SE: 19.3%, RR 0.87, 95%CI 0.70–1.08, *p* = 0.21). Moreover, one-year mortality was comparable in the more recent years (2008–2014) (BE: 17.4% vs. SE: 18.3%, RR 1.05, 95%CI 0.90–1.32, *p* = 0.53) as well as in the most recent years (2015–2018, BE: 8.8% vs. SE: 11.1% RR 1.26, 95%CI 0.89–1.79, *p* = 0.19). Table 2 presents the distribution of valve sizes among BE- and SE-valves.

For hypothesis-generating purposes, valve sizes were divided into three groups: Small (<26 mm, BE: *n* = 759, SE: *n* = 174), medium (26 mm, BE: *n* = 704, SE: *n* = 1249), and large (>26 mm, BE: *n* = 183, SE: *n* = 1118). In the large valve size group, patients receiving BE-valves showed lower mortality than those receiving SE-valves. (BE: 11.1% vs. SE: 17.6%, RR 1.59, 95%CI 1.03–2.44, *p* = 0.03). However, mortality was similar in the medium size group (BE: 13.0% vs. SE: 13.4%, RR 1.03, 95%CI 0.81–1.31, *p* = 0.80) and in the small valve size group (BE: 13.5% vs. SE: 13.1%, RR 0.97, 95%CI 0.64–1.49, *p* = 0.90). Kaplan–Meier curves split by valve size groups are displayed in Figure 3.

### 3.4. One-Year Stroke Rates

In the first year after TAVI, 252 (5.2%) strokes occurred. Of these, 64 (25%) were major, 32 (13%) were minor, 122 (48%) were stroke with undefined severity, and 34 (14%) patients had a TIA. In the propensity-matched population, stroke incidence was 4.9% in patients treated with BE-valves and 5.3% in patients treated with SE-valves (RR 1.09, 95%CI 0.86–1.37, *p* = 0.48). Further, in the unmatched population, stroke incidence was comparable between valve types (BE: 5.4% vs. SE: 5.0%, RR 0.92, 95%CI 0.76–1.11, *p* = 0.38). 

Subgroup analyses by sex, valve generation, valve size, and time period did not show differences in stroke rates between BE- and SE-valves. Men (BE: 4.6% vs. SE: 6.2%, RR 1.36, 95%CI 0.94–1.95, *p* = 0.10) as well as women (BE: 5.2% vs. SE: 4.8%, RR 0.93, 95%CI 0.68–1.26, *p* = 0.62) did not show different stroke rates when split for valve type. Stroke rates were comparable among patients treated with third-generation valves (BE: 3.4% vs. SE: 5.2%, RR 1.51, 95%CI 0.91–2.54, *p* = 0.11). Stroke rates were also similar in patients treated with older-generation valves (BE: 5.1% vs. SE: 5.3%, RR 1.03, 95%CI 0.79–1.35, *p* = 0.84).

Stroke incidence was similar between BE- and SE-valves across different time periods. This was in the earliest years of TAVI (2007–2010, BE: 5.0% vs. SE: 4.4%, RR 0.90, 95%CI 0.54–1.48, *p* = 0.67), in the later years (2011–2014, BE: 4.9% vs. SE: 6.1%, RR 1.26, 95%CI 0.93–1.70, *p* = 0.14), and in the most recent years (2015–2018, BE: 4.9% vs. SE: 4.5%, RR 0.90, 95%CI 0.54–1.51, *p* = 0.70).

Subgroup analyses by valve size did not show significant differences in stroke rates between BE or SE-valves: In the small valve sizes group (BE: 3.4% vs. SE: 4.5%, RR 1.32, 95%CI 0.61–2.87, *p* = 0.48); medium valve size group (BE: 5.1% SE: 3.9%, RR 0.75, 95%CI 0.49–1.14, *p* = 0.18), and in the large valve size group (BE: 3.3% vs. SE: 6.3%, RR 1.92, 95%CI 0.84–4.36, *p* = 0.11). 

### 3.5. Safety of Early Outcomes

Table 3 presents procedural and in-hospital outcomes for patients treated with BE- vs. SE-valves.

Device success, according to VARC2, was lower in patients treated with SE-valves (SE: 91.3% vs. BE: 95.5%, RR 0.96, 95%CI 0.94–0.97, *p* < 0.001). The absence of device success was associated with higher one-year mortality (Odds Ratio (OR) 4.14, 95%CI 3.23–5.30, *p* < 0.001). This association was significant in both the SE-valve patient group (OR 3.69, 95%CI 2.73–4.99, *p* < 0.001) and the BE-valve patient group (OR 5.30, 95%CI 3.41–8.23, *p* < 0.001). Permanent pacemakers were more frequently implanted in patients receiving SE-valves (SE: 20.2% vs. BE: 7.4%, RR 2.73, 95%CI 2.32–3.21, *p* < 0.001). Permanent pacemaker implantation was associated with lower one-year mortality (OR 0.74, 95%CI 0.59–0.92, *p* = 0.008). This association was statistically significant in the BE-valve group (OR 0.59, 95%CI 0.35–0.98, *p* = 0.041), but not in the SE-valve group (OR 0.78, 95%CI 0.61–1.01, *p* = 0.057).

## 4. Discussion

### 4.1. Main Findings

In this large propensity-matched patient study, no differences in one-year mortality and stroke rates between patients treated with BE- or SE-valves were found. In subgroup analyses by sex, valve generation, year of procedure, and valve size, mortality and stroke rates did not differ between patients treated with BE- vs. SE-valves.

### 4.2. Mortality

In >12,000 patients undergoing TAVI, there was no difference in one-year mortality between those treated with BE- vs. SE-valves. To our knowledge, the current study is the largest multinational study comparing BE- and SE-valves.

Our results are in line with five-year follow-up of the CHOICE-trial and one-year follow-up of the SOLVE-trial, which did not find differences in mortality [15,16]. The REPRISE randomized controlled trial found similar two-year mortality, but they compared SE-valves to Lotus valves, which have a different mechanism than BE-valves [9]. Nevertheless, these randomized controlled trials were all underpowered to detect differences in mortality. Our results suggest similar mortality, which is in contrast with two large observational studies that both showed lower mortality in BE- than SE-valves [11,12]. The difference was explained by excess mortality in SE-valve patients during the first three months after TAVI [11]. This is consistent with higher in-hospital mortality found in the current cohort [7]. Nevertheless, in the FRANCE-TAVI registry, the differences remained significant over two years, whereas it was not sustained after discharge in our cohort [7,11]. Despite careful matching, all observational studies involve selection bias. The choice for either a BE- or SE-valve is made by the heart team generally consisting of at least an interventional cardiologist and a cardiothoracic surgeon [17]. The WIN-TAVI registry suggested that patients selected for SE-valve implantation had more comorbidities and higher surgical risk, which is consistent with our results [18]. Another potential unmeasured confounder is operator experience with a particular valve type. Moreover, operators’ learning curve and preference may influence the results. It is impossible to correct for these human variables. The two large observational studies have both been conducted in France [11,12], where local practice may have further induced selection bias. However, we cannot neglect their results suggesting lower mortality in BE-valves. In fact, these contradicting results strengthen the need for an adequately powered randomized study.

Most subgroup analyses did not show different mortality rates between BE- and SE-valves. However, in the large valve size subgroup, BE-valves showed lower mortality than SE-valves. Another observational study found a trend towards increased one-year survival in large BE-valves as compared to SE-valves [19]. In a substudy of the PARTNER-trial, patients receiving large BE-valves showed higher mortality than those receiving small BE-valves [20]. SE-valves have better hemodynamic results than BE-valves due to repositionability and larger aortic orifice areas [16,19]. However, this beneficial effect of SE-valves is more pronounced in small valves [19,21]. Actually, the benefits of BE-valves may outweigh the hemodynamic benefits of SE-valves in larger-sized valves. However, due to the observational nature and different numbers in each valve size category, this finding may be biased.

### 4.3. Stroke 

Non-different stroke rates are consistent with Van Belle and Deharo et al. [11,12] who reported similar stroke incidence in BE and SE-valves at two-year follow-up. In addition, five-year follow-up from the CHOICE trial and two-year follow-up from the REPRISE trial showed similar stroke incidence [9,15]. In the same cohort, we previously showed lower 30-day stroke incidence in patients treated with BE-valves [7]. This effect was not sustained up to one year. In the first days after TAVI, strokes are mostly procedure related [22]. Thereafter, stroke is more associated with atherosclerosis and of trombo-embolic origin [23]. The incidence of new-onset atrial fibrillation, which is strongly associated with stroke, did not differ between patients treated with SE and BE-valves [7,23].

### 4.4. Future Perspectives

Overall, our results did not show significant differences across different valve types used in patients undergoing TAVI, with the exception of larger (>26 mm) valves. However, a certain valve type may be more suitable for an individual patient. Therefore, the decision for a valve type should be tailored to the individual patient. Nowadays there is a heterogeneity of different TAVI devices available with their typical strengths and weaknesses. Future research should identify patient and anatomical characteristics, which improve clinical decision making and valve selection. For instance, female patients with smaller vessels may benefit from SE-valves because of smaller sheath sizes. Moreover, patients with small annuli may benefit from better hemodynamic outcomes in SE-valves [19]. Patients with pre-existing conduction disturbances may benefit from BE-valves with lower pacemaker implantation rates, whereas in patients that already have a permanent pacemaker, there can be preference to treat with SE-valves [7]. If we are able to identify patient characteristics that point towards better outcomes of a certain valve type, we can select the optimal valve for each individual patient. Attempts to individually select the valve type by anatomical and clinical variables have been done in the MIDAS study but this needs to be systematically evaluated [24].

### 4.5. Early Safety Outcomes

Most early outcomes, according to VARC2, were comparable in SE- and BE-valves. However, device success was lower in SE-valves, and device failure was associated with lower mortality in patients treated with SE- as well as BE-valves. Consistent with other studies, patients treated with SE-valves more frequently had a pacemaker implanted. Overall, pacemaker implantation was protective for one-year mortality; however, this association remained significant in BE-valve but not in SE-valve recipients.

### 4.6. Limitations

Although our population was propensity matched, unmeasured variables could have influenced the outcomes of this observational study. Moreover, pre-TAVI echocardiographic features and valve sizes were different between the BE- and SE-valve groups. Post-TAVI echocardiographic measurements were not available. Therefore, we were not able to analyze paravalvular leakage, which has been associated with increased mortality [11]. Additionally, the willingness of principal investigators to collaborate who have their own beliefs on optimal valve therapy could have played a role. Exclusion of certain principal investigators could have invoked selection bias. Numbers of patients in subgroup analyses may not have been large enough to detect statistically significant outcomes. Only patients treated with valves from the CoreValve and Sapien series were included. Therefore, our results are not directly applicable to other valve types.

## 5. Conclusions

Our results suggest that one-year mortality and stroke rates are similar in patients undergoing TAVI with either BE- or SE-valves, with the exception of larger (>26 mm) valves. Since our results add to a number of conflicting studies, our findings highlight the need for a large randomized controlled trial with adequate follow-up.

## Figures and Tables

**Figure 1 jcm-10-04005-f001:**
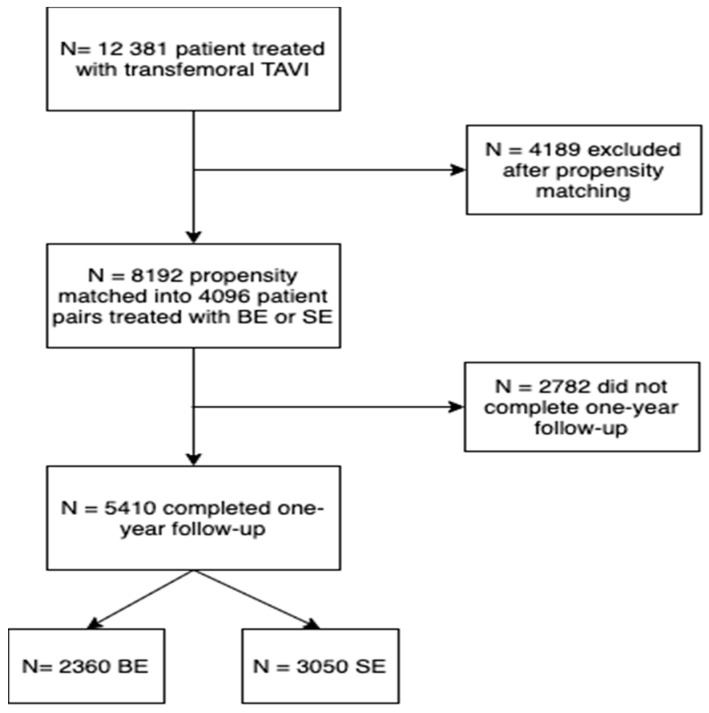
Patient flow chart.

**Figure 2 jcm-10-04005-f002:**
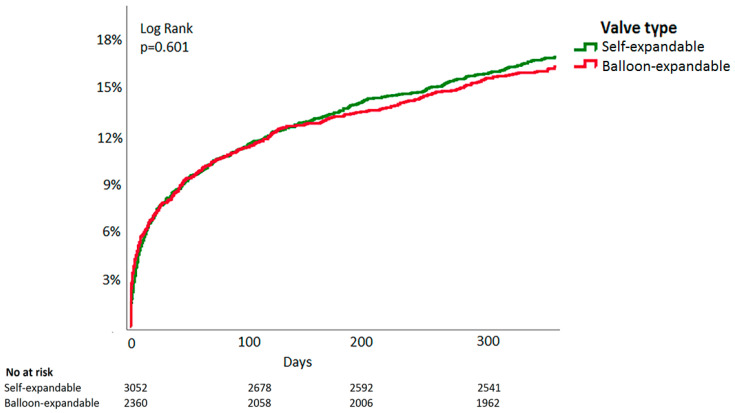
Time-to-event Kaplan–Meier all-cause mortality curve of balloon-expandable vs. self-expandable valves in patients undergoing transcatheter aortic valve implantation.

**Figure 3 jcm-10-04005-f003:**
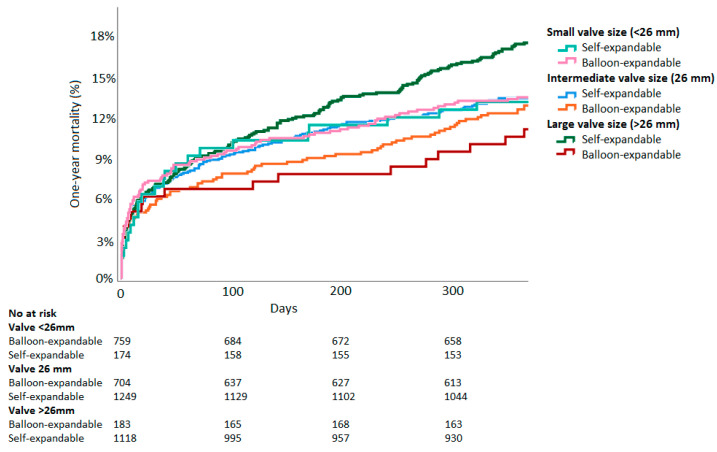
Time-to-event Kaplan–Meier all-cause mortality curves of balloon-expandable vs. self-expandable valves split by valve size.

**Table 1 jcm-10-04005-t001:** Baseline characteristics of the matched patient population included in this study.

	Balloon-Expandable (*n* = 2360)	Self-Expandable (*n* = 3050)	*p*-Value
Age (years)	81.3 ± 7.1	81.3 ± 6.9	0.89
Female gender	1417 (60%)	1822 (60%)	0.82
Body mass index (kg/m^2^)	27.1 ± 4.8	27.1 ± 4.9	0.83
Previous CVA or TIA	236 (10%)	333 (11%)	0.28
Previous MI	317 (13%)	423 (14%)	0.64
Previous PCI	507 (22%)	672 (22%)	0.62
Previous CABG	289 (12%)	379 (12%)	0.84
Diabetes Mellitus	742 (31%)	994 (33%)	0.37
Hypertension	1895 (80%)	2430 (80%)	0.57
Dyslipidaemia	1324 (56%)	1722 (57%)	0.79
Peripheral artery disease	354 (15%)	441 (15%)	0.58
Coronary artery disease	954 (40%)	1230 (40%)	0.94
Atrial Fibrillation	622 (26%)	845 (28%)	0.27
GFR < 30 mL/min	358 (15%)	451 (15%)	0.70
Aortic max gradient (mmHg)	78.2 ± 23.1	80.5 ± 24.3	0.002
Aortic mean gradient (mmHg)	49.2 ± 16.6	50.7 ± 17.1	0.001
Aortic valve area (cm^2^)	0.67 ± 0.20	0.64 ± 0.20	<0.001
Third generation valve	614 (26%)	791 (26%)	0.90
Valve in valve	42 (2%)	90 (3%)	0.002
Valve size (mm)	26 (23–26)	26 (26–29)	<0.001
Year of procedure	2012 (2011–2014)	2012 (2010–2014)	<0.001

CVA = cerebrovascular event. TIA = transient ischemic attack. MI = myocardial infarction. PCI = percutaneous coronary intervention. CABG = coronary artery bypass graft. GFR = Glomerular Filtration Ratio. STS-PROM = Society of Thoracic Surgeons Predicted Risk of Mortality. EuroSCORE = European System for Cardiac Operative Risk Evaluation.

**Table 2 jcm-10-04005-t002:** Distribution of valve sizes in balloon-expandable and self-expandable valves.

	Balloon-Expandable	Self-Expandable
20 mm	4 (0.2%)	-
23 mm	755 (46%)	174 (7%)
26 mm	704 (43%)	1249 (49%)
29 mm	183 (12%)	1022 (40%)
31 mm	-	93 (4%)
34 mm	-	3 (0.1%)

**Table 3 jcm-10-04005-t003:** Procedural and in-hospital outcomes for patients treated with BE- vs. SE-valves.

	Balloon-Expandable (*n* = 2369)	Self-Expandable (*n* = 3050)	Relative Risk (95%CI)	*p*-Value
Conversion to open heart surgery	22 (1.0%)	20 (0.7%)	0.70 (0.39–1.29)	0.25
Device Success	1831 (95.5%)	2161 (91.3%)	0.96 (0.94–0.97)	<0.001
Stroke	67 (2.8%)	94 (3.1%)	1.09 (0.80–1.48)	0.60
Myocardial infarction	23 (1.1%)	22 (0.8%)	0.73 (0.41–1.30)	0.28
Major or life-threatening bleeding	143 (6.9%)	179 (6.5%)	0.95 (0.77–1.18)	0.65
Permanent pacemaker implantation	170 (7.4%)	606 (20.2%)	2.73 (2.32–3.21)	<0.001
Length of hospital stay	7 (5–11)	7 (5–11)	-	0.17

## Data Availability

The data presented in this study are available on request from the corresponding author. The data are not publicly available due to shared copyright of the data.

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
