# Peer review of "Balloon-Expandable versus Self-Expandable Valves in Transcatheter Aortic Valve Implantation: Complications and Outcomes from a Large International Patient Cohort"

_jcm, 2021, doi:10.3390/jcm10174005_

Round 1
Reviewer 1 Report
The paper regards the evaluation of one-year mortality and stroke rates in a large international cohort of patients receiving balloon-expandable versus self-expandable valves. The argument is very actual, since data regarding long term comparison between the two valves are scant. It is a well conducted study, well written, data are clear, convincing and well analyzed. The only major concern is that post intervention echocardiographic data are not available, therefore we have no information at all about perivalvular leak.
Author Response
We would like to thank the editors and reviewers for their thorough review and valuable comments. We have addressed the issues that were raised in detail below.
Open Review
( ) I would not like to sign my review report
(x) I would like to sign my review report
English language and style
|
( ) Extensive editing of English language and style required |
||||
|
|
||||
Yes |
Can be improved |
Must be improved |
Not applicable |
||
Does the introduction provide sufficient background and include all relevant references? |
(x) |
( ) |
( ) |
( ) |
|
Is the research design appropriate? |
(x) |
( ) |
( ) |
( ) |
|
Are the methods adequately described? |
(x) |
( ) |
( ) |
( ) |
|
Are the results clearly presented? |
(x) |
( ) |
( ) |
( ) |
|
Are the conclusions supported by the results? |
(x) |
( ) |
( ) |
( ) |
|
The paper regards the evaluation of one-year mortality and stroke rates in a large international cohort of patients receiving balloon-expandable versus self-expandable valves. The argument is very actual, since data regarding long term comparison between the two valves are scant. It is a well conducted study, well written, data are clear, convincing and well analyzed. The only major concern is that post intervention echocardiographic data are not available, therefore we have no information at all about perivalvular leak.
We agree with the reviewer that details on paravalvular leakage would be an interesting addition to this analysis. In fact, paravalvular leakage has been associated with increased mortality in previous studies. Unfortunately, post intervention echocardiographic data were not available. Therefore we have stated this in the limitations section of the discussion.
4.6 Limitations lines 329-331
Post-TAVI echocardiographic measurements were not available. Therefore we were not able to analyse paravalvular leakage, which has been associated with increased mortality [11].
Reviewer 2 Report
I have read with interest this paper entitled “Balloon-expandable versus self-expandable valves in transcatheter aortic valve implantation: complications and outcomes from a large international patient cohort”.
In this manuscript, the authors compared the rates of 1 year mortality and stroke between patients with BE and SE. As a result, the authors demonstrated that 1 year mortality and stroke rates were comparable in patients with severe AS undergoing TAVI with either BE or SE.
Overall, I think that this article has well designed and written.
However, there are some issues which should be resolved and clarified.
Comments
There may be a significant selection bias in the selection of these valves.
As the authors describe in the discussion part, the surgeon selects these valves according to the anatomical findings of the valves.
In other words, the location and degree of calcification and small annulus diameter are important information.
Authors should clarify in supplementary material about the anatomical information by CT measurement and how the valve was selected in each registry
In addition, the information on cardiac function or coronary artery disease is also important because it is necessary to withstand rapid pacing during procedure in BE group.
Therefore, the authors should clarify such information.
Author Response
We would like to thank the editors and reviewers for their thorough review and valuable comments. We have addressed the issues that were raised in detail below.
Open Review
(x) I would not like to sign my review report
( ) I would like to sign my review report
English language and style
( ) Extensive editing of English language and style required
( ) Moderate English changes required
(x) English language and style are fine/minor spell check required
( ) I don't feel qualified to judge about the English language and style
|
|||||
Yes |
Can be improved |
Must be improved |
Not applicable |
||
Does the introduction provide sufficient background and include all relevant references? |
(x) |
( ) |
( ) |
( ) |
|
Is the research design appropriate? |
( ) |
(x) |
( ) |
( ) |
|
Are the methods adequately described? |
( ) |
(x) |
( ) |
( ) |
|
Are the results clearly presented? |
(x) |
( ) |
( ) |
( ) |
|
Are the conclusions supported by the results? |
( ) |
( ) |
( ) |
( ) |
|
Comments and Suggestions for Authors
I have read with interest this paper entitled “Balloon-expandable versus self-expandable valves in transcatheter aortic valve implantation: complications and outcomes from a large international patient cohort”.
In this manuscript, the authors compared the rates of 1 year mortality and stroke between patients with BE and SE. As a result, the authors demonstrated that 1 year mortality and stroke rates were comparable in patients with severe AS undergoing TAVI with either BE or SE.
Overall, I think that this article has well designed and written.
However, there are some issues which should be resolved and clarified.
Comments
There may be a significant selection bias in the selection of these valves.
As the authors describe in the discussion part, the surgeon selects these valves according to the anatomical findings of the valves.
In other words, the location and degree of calcification and small annulus diameter are important information.
Authors should clarify in supplementary material about the anatomical information by CT measurement and how the valve was selected in each registry
Indeed the degree of calcification could be an unmeasured confounder in the valve type selection process. Therefore we have added Supplementary Table S1. This table presents an overview of the original studies, number of included patients, proportion of balloon- and self-expandable valves, procedure of valve selection and CT measurement of aortic valve calcification. As presented here, all studies selected valve types using a heart team based approach, typically consisting of at least an interventional cardiologist and a cardiac surgeon. This is reassuring because then valve choice cannot be based on just the personal preference of one operator. If available, we have mentioned the annulus diameter, porcelain aorta and aortic calcification.
2.1 Study design and patient sample, lines 82-84
The choice for a BE or SE valve type was made by the local heart team of each participating center. An overview of the included studies is presented in Supplementary Table S1.
Study name and PMID |
CENTER* (n=) |
BE vs SE* (%)
|
Valve selection |
Calcification on CT |
Brazilian TAVI registry 27496637 |
768 |
BE: 195 (25%) SE: 573 (75%) |
Heart team in different centers |
Not available |
FRANCE-2 25240554 |
2347 |
BE: 1506 (64%) SE: 841 (36%) |
Heart team in different centers |
8% porcelain aorta |
Milano 27184169 |
515 |
BE: 270 (52%) SE: 245 (48%) |
Multidisciplinary heart team |
AV calcification: 52.6% grade 1-2, 46.6% grade 3-4. Annulus diameter 23.0 mm. |
Verona 27621826 |
346 |
BE: 254 (73%) SE: 92 (27%) |
Institutional Heart Team |
Not available |
Rabin 27726854 |
544 |
BE: 120 (22%) SE: 424 (78%) |
Institutional Heart Team |
Not available |
Padova 26603025 |
447 |
BE: 352 (79%) SE: 95 (21%) |
Institutional Heart Team |
16.3% porcelain aorta |
Spanish TAVI registry 24774108 |
5320 |
BE: 2451 (46%) SE: 2869 (54%) |
Heart team in different centers |
Annulus diameter 23.2 mm |
BRAVO-3 26477635 |
732 |
BE: 500 (68%) SE: 232 (32%) |
Institutional Heart Team |
Not available |
WIN-TAVI 27491609 |
785 |
BE: 366 (47%) SE: 419 (53%) |
Institutional Heart Team |
Not available |
OBSERVANT 26271063 |
577 |
BE: 225 (39%) BE: 352 (61%) |
Heart team in different centers |
Patients with porcelain aorta were excluded. Annulus diameter 22.2 mm. |
Supplementary Table S1. Overview of original studies, number of patients, proportion of balloon- and self-expandable valves, procedure of valve selection and aortic valve calcification as measured by computed tomography.
* number of patients from the original study included in the CENTER database. CT = computed tomography.
In addition, the information on cardiac function or coronary artery disease is also important because it is necessary to withstand rapid pacing during procedure in BE group.
Therefore, the authors should clarify such information.
The presence of coronary artery disease, history of myocardial infarction and treatment with percutaneous coronary intervention were included in the propensity matching score. As presented in table 1, these variables were equally distributed across BE and SE-valve recipients. Moreover, the history of a coronary artery bypass graft was equal in both groups. We have elaborated this further in the manuscript.
3.2. Baseline characteristics of the matched study population, lines 170-172
In particular, the presence of coronary artery disease was comparable.